# The Removal of a Textile Dye from an Aqueous Solution Using a Biocomposite Adsorbent

**DOI:** 10.3390/polym14122396

**Published:** 2022-06-13

**Authors:** Hana Ferkous, Karima Rouibah, Nour-El-Houda Hammoudi, Manawwer Alam, Chahrazed Djilani, Amel Delimi, Omar Laraba, Krishna Kumar Yadav, Hyun-Jo Ahn, Byong-Hun Jeon, Yacine Benguerba

**Affiliations:** 1Laboratoire de Génie Mécanique et Matériaux, Faculté de Technologie, Université de 20 Août 1955, Skikda 21000, Algeria; hanaferkous@gmail.com (H.F.); a_delimi03@yahoo.fr (A.D.); 2Département de Technologie, Université de 20 Août 1955 de Skikda, Skikda 21000, Algeria; chahrazed_dj@yahoo.fr; 3Laboratory of Materials-Elaboration-Properties-Applications (LMEPA), University of MSBY Jijel, PB98 Ouled Aissa, Jijel 18000, Algeria; karima.rouibah@univ-jijel.dz (K.R.); invincible3089@gmail.com (O.L.); 4Department of Engineering Proceeding, Faculty of Sciences and Technology, University MSBY Jijel, PB98 Ouled Aissa, Jijel 18000, Algeria; 5Laboratoire de Biopharmacie et Pharmacotechnie, Ferhat Abbas Setif 1 University, Setif 19000, Algeria; hammoudi_nourelhouda@yahoo.com; 6Department of Chemistry, College of Science, King Saud University, P.O. Box 2455, Riyadh 11451, Saudi Arabia; maalam@ksu.edu.sa; 7Laboratoire LRPCSI, University of 20 Août 1955, El Hadaiek Road, Skikda 21000, Algeria; 8Faculty of Science and Technology, Madhyanchal Professional University, Ratibad, Bhopal 462044, India; envirokrishna@gmail.com; 9Department of Earth Resources and Environmental Engineering, Hanyang University, Seoul 04763, Korea; hjahn93@hanyang.ac.kr

**Keywords:** adsorption, DFT, Monte Carlo simulations, Langmuir–Freundlich

## Abstract

The adsorption mechanisms of methylene blue (MB) onto olive waste (residue) treated with KOH (OR-KOH) and onto an OR-KOH and PEG–silica gel composite (OR-KOH/PEG-SG) at various temperatures were investigated using a combination of experimental analysis and Monte Carlo ab-initio simulations. The effects of adsorption process variables such as pH, temperature, and starting adsorbate concentration were investigated. The experimental data were fitted to Langmuir and Freundlich models. The maximum adsorption capacities of MB onto OR-KOH and OR-KOH/PEG-SG adsorbents reached values of 504.9 mg/g and 161.44 mg/g, respectively. The experimental FT-IR spectra indicated that electrostatic attraction and hydrogen bond formation were critical for MB adsorption onto the adsorbents generated from olive waste. The energetic analyses performed using Monte Carlo atomistic simulations explained the experimental results of a differential affinity for the investigated adsorbents and confirmed the nature of the interactions between methylene blue and the adsorbents to be van der Waals electrostatic forces.

## 1. Introduction

To the best of our knowledge, the first use of natural dyes was reported as far back as 2600 BC but only in 1856, when William Henry Perkin attempted to synthesize artificial quinine from allyltoluidine to treat malaria, was the first dye material synthesized which he called “mauve” (aniline, a basic dye). After that, the synthetic dye industry was born [1].

Nowadays, synthetic dyes represent a relatively large group of organic chemical compounds used in our daily life [2,3]. Global production is estimated to be about 700,000 tons/year, of which 140,000 tons are released into effluents during various applications and manufacturing stages because of wrong or negligent discharges [4,5]. If not adequately treated, these effluents, composed of surfactants, biocide compounds, solid suspensions, dispersal and mooring agents, dyes, and metal traces, are toxic to most living organisms [6,7]. Their heterogeneous composition makes it difficult to reach pollution levels less or equal to those imposed by environmental standards when adopting the traditional treatments commonly used in municipal wastewater plants [8]. 

One of the primary concerns of water pollution is the dye contamination of wastewater from the textile industry, which is a significant chemical, physical, and aesthetical pollutant [9]. Eutrophication and disturbance of aquatic life (e.g., limiting access to sunlight and oxygen) presents a potential environmental danger. Possible bioaccumulation also represents a further threat, affecting human health and the environment. 

Several techniques can be used for pollution treatment, and adsorption has been commonly considered a reliable, versatile, and efficient option [10]. When applying this technique, the adsorbents that have been most adopted for use are commercial activated carbons, which are usually expensive despite their proven efficiency. In some cases, their use produces delayed pollution, representing an additional environmental threat [11,12]. Recently, many adsorbents have been reported by researchers in the context of dye removal from wastewaters, such as agricultural wastes [13,14,15], super-absorbent hydrogels [16], chitosan-based adsorbents [17], Ba/Co-PEG nanocomposites [18], PEG-Crosslinked biomass polymers [19], and silica-PEG intercalated clay hybrids [20]. Using agricultural waste as an eco-friendly precursor would provide a valid alternative, assuring a lower economic and polluting impact [21,22]. Moreover, this is a valuable operation that simultaneously includes the disposal of waste material, contributing to the production of added-value material and economic savings, which should be promoted as part of the circular economy framework.

Although many different agricultural wastes have been proposed as porous adsorbents, few activation treatments have been tested and applied to the reuse of olive wastes. Creating a new synthesis route for the realization of a new adsorbent can be time and energy-consuming if the end absorbent performance is inadequate. Significant help can be derived from atomistic simulations such as those based on the Monte Carlo method to initially assess the suitability of a certain adsorbent for capturing a specific compound. This simulation at the atomistic level is a powerful tool for the energetic evaluation of the adsorption pseudoreaction and the determination of the properties of the bonds that occur upon adsorption. It could be used to extract information about the factors that have a considerable effect and, subsequently, those that can increase any effects, starting with the chemical properties of the pseudoreactants (i.e., adsorbent and adsorbate). The possible relations between adsorbent and adsorbate and the determination of the energy involved could provide helpful information about the adsorption and magnitude of the adsorption capacity. The simulation of adsorption processes, based on the possibility of an experimental check of the retrieved results, is an innovative tool in the field of adsorption research, which also allows for a quick and easy evaluation of the adsorption properties of a stated system, helping to understand its potential for real application.

A novel adsorbent (OR-KOH/PEG-SG sample) was created by treating olive waste (residue) with KOH and then mixing it with a PEG-silica gel composite. The removal of methylene blue (MB) from water, using olive waste activated with KOH (OR-KOH sample) and OR-KOH/PEG-SG support, was examined under varying temperatures, pH, and starting adsorbate concentrations. The experimental findings were interpreted and supported by molecular simulations using a classical modeling approach. Molecular simulations were used in combination with fundamental data analysis to better study the adsorption process. Furthermore, grand canonical Monte Carlo (GCMC) atomistic simulations were used to investigate the microscopic parameters that led to the dye binding to the two adsorbents. Finally, a comparison of the obtained data was performed to gauge the usefulness of this new approach for research.

## 2. Materials and Methods

### 2.1. Preparation of Adsorbents

The preparation of the low-cost olive waste adsorbents was passed through different steps. Firstly, a porcelain mortar was used to reduce the grain size of the raw material down to a relatively fine particle size of a few μm. A sample of 100 g of the crushed olive waste was then immersed and stirred at 700 rpm in 1 L of hot water to remove any dust, oil, or any other adhesive impurities and water-soluble substances until clear rinsing water was obtained. The sample was then centrifuged, filtrated, and dried in a vacuum oven at 50 °C for 24 h. The total amount retrieved was 86 g from the original 100 g. After purifying the olive residue, it was dried and crushed to obtain a fine powder with a particle size smaller than 500 µm. It then underwent a chemical activation with a basic solution of KOH (0.2 M). An amount of 4.48 g of KOH was dissolved in 400 mL of distilled water. The mixture (16 g of waste olive powder + KOH basic solution) was stirred for 30 min at room temperature, filtrated, and then dried; firstly at 50 °C for 24 h and then the temperature was increased to 120 °C for 1 h 30 min. After that, the recovered sample was washed using distilled water to eliminate the excess activating agent (KOH) and the soluble fractions of ashes. The rinsing water was then treated with AgNO_3_. If this water remained clear, the washing was stopped. Otherwise, the same rinsing operation is repeated. Finally, the cleaned sample was dried at 50 °C for 24 h.

A mixture of Si (OH)_4_, distilled water, and ethanol (molar ratio 1:10:5, respectively) was stirred for 30 min to prepare the PEG–silica gel composite. Ethanol was used as a cosolvent and homogenizing medium [23,24] to reduce Si(OH)_4_ in water. Then, a hydrochloric acid solution ([HCl] = 0.611 mol/L, V = 8 mL) was added to the mixture, and the temperature was increased up to 50 °C. After 10 min, a molten amount of PEG3000 was added into the mix, and after a further 20 min, a sodium carbonate solution ([Na_2_CO_3_] = 1.21 mol/L, V = 4 mL) was also added. The gel was formed immediately. Finally, the product was dried in a vacuum oven at 50 °C for 24 h, and the desired composite PEG–silica gel (70/30 weight) was obtained. Equal amounts of OR-KOH adsorbent and PEG-SG (mass ratio 1:1) were added to 30 mL of distilled water to improve the olive waste adsorption capacity. The mixture was stirred for 24 h at ambient temperature, filtrated, and then dried in a vacuum oven at 50 °C for 24 h so as to obtain the adsorbent (OR-KOH/PEG-SG sample).

### 2.2. Chemicals

The dye investigated in this work was methylene blue (MB: C_16_H_18_N_3_SC_1_). Potassium hydroxide (KOH) was used for the chemical activation of the adsorbent. Both NaOH and HCl solutions were used to adjust the pH values during adsorption tests. The composite was prepared using silica gel (Si(OH)_4_). Sodium carbonate Na_2_CO_3_ served as a catalyst. The use of silver nitrate AgNO_3_ allowed for the verifying of the clearness of rinsing water. Ethanol and distilled water were utilized as solvents. Finally, polyethylene glycol (PEG, C_2_H_4_O), also called macrogol in the medical domain (molar weight = 3000 g/mol), was used to support the dispersion of the adsorbent. All the chemicals were purchased from Sigma–Aldrich as reagent grade.

### 2.3. Characterization of OR-KOH and OR-KOH/PEG-SG Adsorbents

The adsorbents were characterized in terms of point of zero charge $pH_{PZC}$. To this aim, 20 mL of distilled water at neutral pH was added to 12 Erlenmeyer flasks of 50 mL. Then, the pH of the 12 solutions was adjusted with either HCl (0.1 M) or NaOH (0.1 M). After that, 20 mg of the adsorbent powder was added to all the solutions. Each suspension was stirred for 24 h, and the final pH values were measured using a pH meter (OHAUS). The final pH minus the initial pH value, as a function of the initial pH values, was plotted to obtain *pH_PZC_*. The *pH_PZC_* is calculated as the terminal pH value corresponding to the intersection of the obtained curve with the abscissa axis.

The infrared spectra of the adsorbents before and after modification and the PEG-SG were recorded by an FTIR spectrophotometer (Shimadzu) in the range of 4500–500 cm^−1^. 

### 2.4. Batch Adsorption Experiments

Adsorption isotherms were determined using 10 Erlenmeyer flasks (50 mL) filled with 20 mL of aqueous solutions with different MB concentrations (from 25 to 800 mg/L). Then, 20 mg of either OR-KOH or OR-KOH/PEG-SG adsorbent was added to each solution, and the batch flasks were stored in an oven at controlled temperatures (30, 40, and 50 °C). The batch flasks were stirred continuously at 200 rpm and stored in an oven at controlled temperatures (30, 40, and 50 °C). At the end of the experiment, the solutions were centrifuged for 30 min at 6000 rpm.

The adsorption tests were repeated twice to assess the reproducibility of the results, with the average values considered. 

MB concentrations were measured using a UV–Visible spectrophotometer (SP-3000 nano OPTIMA) at a maximum absorption wavelength of λ_max_ = 664 nm. UV–Visible spectrophotometry is a quantitative analysis technique that measures a given chemical substance’s abundance or optical density in a solution based on the Beer–Lambert law.

The adsorption capacity of BM was calculated using the following relation:(1)$$q_{e}=\frac{(C_{0}-C_{e})\cdot V}{m}$$
where $q_{e}$ is the amount of adsorbed BM at equilibrium (mg/g), *V* is the volume of the solution (L), *C*_0_ and *C_e_* are the initial and equilibrium dye concentrations (mg/L), respectively, and *m* is the amount of the adsorbent (g).

MB adsorbed quantities can also be estimated by R (%), which is the BM removal percentage defined by the following equation [25]:(2)$$R\;\left(\%\right)=\frac{\left(C_{0}-C_{e}\right)}{C_{0}}\times 100$$

The effect of pH on the MB adsorption process was investigated at ambient temperature by adjusting the solution pH using the appropriate amount of either HCl or NaOH. After equilibrium, the adsorbent was separated by filtration, and the filtrate was analyzed by UV spectrophotometry.

### 2.5. Adsorption Isotherms Analysis 

Modeling analysis was carried out by considering the classical adsorption models for a basic interpretation of the experimental data and supporting the subsequent process simulation.

The following expression gives the Langmuir adsorption model:(3)$$q_{e}=q_{max}\frac{K_{L}C_{e}}{1+K_{L}C_{e}}$$
where the dye adsorption capacity is $q_{e}$ (mg/g), the equilibrium dye concentration in the liquid is $C_{e}$ (mg/L), the maximum adsorption capacity is $q_{max}$ (mg/L), and the Langmuir constant is $K_{L}$ (L/mg) which determines the extent of the interaction between the adsorbate and the surface. 

The Freundlich model is given as follows:
(4)$$q_{e}=K_{f}{\left(C_{e}\right)}^{1/n}$$
where $q_{e}$ is the dye adsorption capacity, $K_{f}$ ((mg/L)^(1/*n*)^) is the Freundlich constant, and 1/n is the adsorption intensity, characterizing the heterogeneity of the system [26]. 

### 2.6. Computational Study

Adsorption systems were simulated by adopting the Metropolis Monte Carlo simulations (MCS) [27] using the adsorption locator module [28] in Materials Studio 2017™. The MCS process is used to find the adsorbate/adsorbent configuration, corresponding to the lowest total energy. In an MCS framework, the equilibrium conditions can be predicted starting from the chemical potential of fluid and solid phases, accounting for the adsorbent/adsorbate structures with properties at an atomistic level. Simulations were carried out for both the adsorption systems: MB/OR-KOH and MB/OR-KOH/PEG-SG. Two different boxes were constructed: (13.6 × 13.6 × 46.50 Å^3^) and (25.9 × 25.9 × 25.9 Å^3^), respectively. The created vacuum ran along the Cz-axis with periodic boundary conditions to model a representative part of the interface, devoid of arbitrary boundary effects. For calculating the interaction forces in the whole simulation procedure, the DREIDING force field [29] was implemented.

## 3. Results and Discussion

### 3.1. Point of Zero Charge pH (${pH}_{PZC}$)

The pH of a diluted solution is an essential factor for determining the adsorption properties of a given adsorbent. In turn, the behavior of an adsorption system is greatly influenced by the $pH_{PZC}$ (the pH at which OH^−^ and H^+^ ions are adsorbed to the same extent). The presence of OH^−^ and H^+^ ions in the solution can change the surface charge potential of adsorbents. In Figure 1, the zero charge point values are given by the intersection of the horizontal line y = 0 and the curve $pH_{f}-pH_{i}=f\;\left(pH_{i}\right)$.

OR-KOH/PEG-SG and OR-KOH show zero charge point values around 7.2 and 6.6, respectively. This $pH_{PZC}$ = $pH_{i}$ point divides the adsorbent area into two subsequent areas:
✓In the range of lower values ($pH_{i}$ < $pH_{PZC}$), the adsorbent surface will be protonated due to an excess of H^+^. The positively charged adsorbents are more attractive to negatively charged compounds.✓In the range of higher values ($pH_{i}>pH_{PZC}$), the adsorbent surface will be deprotonated by the presence of OH^−^ ions in the solution. Adsorbents in this pH range are more attractive to positively charged compounds [30].

Since MB is a cationic dye, it was expected that the best adsorption performance would be obtained for $pH_{i}>pH_{PZC}$.

### 3.2. Fourier Transform Infrared Spectrometry (FTIR)

The olive residue consists of epidermal cells that contain cellulose, hemicellulose, and lignin. The latter contains polar functional groups such as alcohols, aldehydes, ketones, carboxylic, phenolic, and others [31]. Once the adsorbent is synthesized, these groups form active sites on the material surface. The FTIR spectra of the adsorbents, their raw precursor (OR, olive residues), and the PEG-SG used for the synthesis are shown in Figure 2.

The OR-KOH/PEG-SG sample spectrum showed characteristics close to the OR-KOH precursor spectrum, with slight changes. Both spectra show similar peaks but slightly differ in intensity (see Figure 2).

The spectra of all the materials showed a hydroxyl characteristic band at 3440–3442 cm^−l^, while the band at 2700–2900 cm^−1^ was attributed to the presence of =C-H stretching. This peak was more intense for the OR solid, while the subsequent treatments with KOH tended to progressively reduce it (see the peaks of OR-KOH and OR-KOH/PEG-SG). The peak at 1720 cm^−1^ corresponds to the aromatic C=0 stretching. The phenol -OH is characterized by the peak at 1340 cm^−1^ (OR spectrum) which was not visible for the two materials OR-KOH and OR-KOH/PEG-SG due to KOH treatment. The aromatic ethers group =C-O- (present in OR, OR-KOH, and OR-KOH/PEG-SG spectra) is characterized by a peak at 1000–1300 cm^−1^. The ether group of PEG was detected at 1240 cm^−1^. The band at 806 cm^−1^ might correspond to either aliphatic or aromatic C-H. Additionally, the existence of C-X bonding could explain the strong peak observed at 478 cm^−1^.

Lower intensities were noticed for the PEG-SG peaks recorded at 3341 cm^−1^, 2917 cm^−1^, 1340 cm^−1^ (corresponding to OH groups, C-H aliphatic groups, and C=O, respectively), in comparison to those of raw olive residue samples. This may indicate a reduction in these groups, strengthening the hypothesis that they represent the active sites where bonds form, leading to the further disappearance of these functional groups.

### 3.3. Effect of Initial pH on MB Adsorption 

The pH of a solution is a significant parameter, especially when coupled with the pH_PZC,_ because different electrostatic phenomena can arise during adsorption interactions, significantly affecting the adsorption capacity [32]. pH may cause a change in the adsorbent’s surface charge, the adsorbate’s ionization degree, and the degree of dissociation of functional groups of both adsorbent active sites and adsorbate molecule [33,34]. The effects of pH on MB adsorption onto OR-KOH and OR-KOG/PEG-SG adsorbents is illustrated in Figure 3.

Figure 3 shows that strongly acidic solutions exhibit poor removal efficiency. The higher the pH, the more efficient the MB removal is from an aqueous solution. Even if OR-KOH/PEG-SG shows a lower value of adsorption capacity, it is more sensitive to pH variation than OR-KOH, as evidenced by the obtained values of *R*%, which were primarily superior to those shown by OR-KOH/PEG-SG, particularly in the range of 3 < pH < 10. It is to be noted that, for the OR-KOH samples, the increase in the removal efficiency (R) was very high until 6.2 (<$pH_{PZC}$), where a maximal removal rate of about 90% was reached; thereafter, the curve exhibits an almost constant quality and a result of about 87% removal at pH 12. On the contrary, the OR-KOH/PEG-SG curve tends to vary less sharply in a quasi-linear trend, achieving a maximal *R*% value of 90% at pH 11.

It is worth noting that the maximal removal efficiency (*R*%) reached for OR-KOH is 90% at pH 8, whereas the maximal value (*R*%) achieved by OR-KOH/PEG-SG occurred at pH 12.
The surface charge below $pH_{PZC}$ _(_pH < $pH_{PZC}$) is positive, whereas this surface carries negative charges when pH > $pH_{PZC}$. Therefore, the high elimination percentage in high pH values may be caused by an electrostatic attraction between negatively charged material and positively charged MB.For low pH, a decrease in the elimination percentage was noticed due to the electrostatic repulsion between positively charged material and MB cations. Additionally, the competition with the H^+^ cations weakens the adsorption capacity [35].

In the following, the pH of the solutions is set at pH = 6, which is the natural pH of the methylene blue solution.

### 3.4. Adsorption Isotherms

The adsorption isotherms of MB onto OR-KOH and OR-KOH-PEG/SG adsorbents, measured at different temperatures (30, 40, and 50 °C) and using a wide range of initial concentrations, are shown in Figure 4 and Figure 5.

An increase in temperature determined the adsorption capacity for both adsorbents, which was more evident when the equilibrium concentration was higher. This evidence could be ascribed to the particular affinity of both adsorbents towards the water due to the presence of OH functional groups from hydrogen bonds (Figure 2). Consequently, an increase in the temperature results in a nonnegligible decrease in the water adsorbed. In turn, this decrease can favor the increase in the adsorption of MB, which overall appears as endothermic adsorption. However, the trend of adsorption isotherms was significantly different for the two adsorbents. In fact, for OR-KOH, the adsorption capacity gradually increased with equilibrium concentration, indicating the gradual saturation of the solid and an increasing interaction between MB and OR-KOH due to the exploitation of different adsorption sites. This would likely result in a homogeneous surface of the adsorbent being observed and a gradual increase of MB adsorption capacity with equilibrium concentration. 

On the contrary, for OR-KOH/PEG-SG, a rapid increase in adsorption capacity was observed for a very low value of equilibrium concentration. The curves tend to plateau at high equilibrium dye concentrations, which was not the case for the tests carried out with OR-KOH. Thus, we hypothesize the existence of higher energy adsorption sites that were initially exploited, which are characterized by a higher degree of variability for the OR-KOH/PEG-SG adsorbent.

A modeling analysis was carried out based on Langmuir and Freundlich models, expressed by Equations (3) and (4), to further analyze the experimental data. The objective was to determine which models could best interpret the experimental data by comparing R^2^ (correlation coefficient) and RMSE (root mean square error) derived from the analytical fitting.

The RMSE estimates the standard deviation of the random error for the two models considered in this work. A lower RMSE corresponds to a better fitting model. The RMSE was calculated by Equation (5):(5)$$RMSE=\sqrt{\frac{1}{n-p}\sum_{1}^{n}{\left(q_{e,exp}-q_{e,calc}\right)}^{2}}$$
where *n* is the number of experimental data points, *p* is the number of parameters in the isotherm model, and *q_e(exp)_* (mg/g) and *q_e(calc)_* (mg/g) are the experimental and calculated values of adsorption capacity in equilibrium, respectively.

Table 1 shows the results of the isotherm data modeling. For T = 30 °C, the experimental data were better interpreted by the Langmuir model for both adsorbents. The values of RMSE are 8.065 and 3.161 for the OR-KOH and OR-KOH/PEG-SG adsorbents, respectively. The Freundlich isotherm was the best model for the data obtained for OR-KOH at higher temperatures. For OR-KOH/PEG-SG, the Langmuir model gives better results than the Freundlich model at 40 °C, but when T > 40°, the Freundlich model is the best for correlating the experimental data. From the modeling data, it can be confirmed that MB adsorption was higher onto OR-KOH than OR-KOH/PEG-SG. This is probably because the polymer acts as a barrier that prevents the passage of MB molecules to the adsorption sites. Furthermore, in the Freundlich adsorption isotherm, the magnitude of *n* indicates the favourability of adsorption; when *n* is in the range of 2–10, the adsorption is excellent and poor if *n* is less than 1 [36,37]. The value of *n* is greater than 2 for OR-KOH/PEG-SG, suggesting that MB is more strongly adsorbed onto OR-KOH/PEG-SG.

Adsorption at low temperatures is monolayer and occurs at specific homogeneous adsorbent sites. In these conditions, the competition with water is probably intense, causing surface saturation of the adsorbents to occur. For higher temperatures, the endothermicity of the adsorption mechanism helps capture more adsorbate molecules. The adsorption may turn multilayer, probably because of new sites becoming available for MB adsorption.

### 3.5. Comparison with Other Adsorbents

The maximum sorption capacity of the OR-KOH and OR-KOH/PEG-SG was compared with the results reported in recent work on MB removal, detailed in Table 2. The olive waste activated with KOH (OR-KOH) exhibits the highest adsorption capacity. The bio-based adsorbent OR-KOH/PEG-SG demonstrated greater adsorption of the dye when compared to activated carbon obtained from banana stem [38], Silica–PEG intercalated clay hybrids [20], Fe_3_O_4_/biochar nanocomposite [39], chitosan/laterite/iron oxide-based bio-composite [40] and PEG-Crosslinked β-CD polymers [19].

### 3.6. Monte Carlo Computational Study

Monte Carlo atomistic simulations were carried out to deepen our theoretical investigations. The adsorption locator’s simulated annealing (SA) optimization method allows for specific parameters that control the simulated annealing temperature cycle and define the relative probabilities (ratios) of the different types of Monte Carlo steps. The adsorption energy is defined as the energy released during the adsorption of MB onto OR-KOH or MB/OR-KOH/PEG-SG. It is the sum of the rigid adsorption (unrelaxed adsorbate) and deformation (for relaxed adsorbate) energies. The term (dE_ad_/dN_i_) accounts for the energy needed to remove the adsorbate from the surface. Higher negative adsorption energy values indicate a more stabilized and robust interaction between MB and the adsorbents. The results reported in Table 3 show the different energies obtained for the MB/OR-KOH and MB/OR-KOH/PEG-SG adsorption systems. Table 2 shows that MB is more strongly adsorbed on OR-KOH/PEG-SG than it is on OR-KOH (adsorption energy), showing consistency with the experimental and modeling results (cf. Langmuir and Freundlich adsorption constants, Table 1). The regeneration of the OR-KOH adsorbent requires less energy to desorb MB with −14.498 kcal mol^−1^ compared with −24.565 kcal mol^−1^ for OR-KOH/PEG-SG. In addition, for water desorption, a higher amount of energy is required for the OR-KOH adsorbent, which accounts for the higher increment in MB adsorption capacity with the maximum investigated temperature.

Figure 6 shows that the structure of OR-KOH is less sterically hindered than the structure of OR-KOH/PEG-SG. This can be explained by the fact that PEG-SG plays the role of a barrier (at a macroscopic scale), thus hindering the passage of MB towards the adsorption sites (OR-KOH). This may partially explain why MB is less adsorbed on OR-KOH/PEG-SG. It should be noted that MB must pass through all of this congestion to arrive at the OR-KOH sites (Lignin). Energy distributions, such as total energy, average total energy, van der Waals energy, electrostatic energy, and intermolecular energy (plotted after 15,000 steps per cycle of adsorption locator module) for MB/OR-KOH and MB/OR-KOH-PEG-SG systems, are presented in Figure 7. The dominant form of interaction energy for the two adsorbents was the van der Waals energy.

## 4. Conclusions

In this work, the adsorption of methylene blue onto two different adsorbents derived from olive residues, namely OR-KOH (treated with KOH) and OR-KOH-PEG (after further treatment with PEG-SG), was experimentally investigated. MB was found to adsorb better onto OR-KOH/PEG-SG at low temperatures (30 °C), whereas KOH (olive residue treated by KOH (0.2 M)) produced better results at higher temperatures (40 °C and 50 °C). For both the adsorbents, an increase in temperature produced a significant increase in MB adsorption capacity, likely due to a reduction in water adsorption. The FTIR characterization data confirmed the affinity of the adsorbents towards the water by hydrogen bonding. MB adsorption capacity also increased with solution pH, and for the OR-KOH/PEG-SG adsorbent, a higher pH value was necessary to grant the maximum adsorption capacity, in line with the results derived from $pH_{PZC}$ investigation.

Monte Carlo atomistic simulations were carried out to support the experimental investigations, along with thorough energy characterizations of MB adsorption onto the two adsorbents in an aqueous solution. The results indicated that MB is more strongly adsorbed on OR-KOH/PEG-SG than on OR-KOH, consistent with the experimental and modeling results. van der Waals electrostatic forces are those involved in the adsorption phenomena. The simulations also showed that the presence of PEG-SG exerts a hindrance effect on MB molecules, thus explaining the reduction in adsorption capacity.

This work has demonstrated the possibility of integrating two different methodologies (experimental and theoretical–simulative) as an innovative tool for investigating newly synthesized adsorbents and for determining the energetic interactions that can significantly influence adsorption capacity.

## Figures and Tables

**Figure 1 polymers-14-02396-f001:**
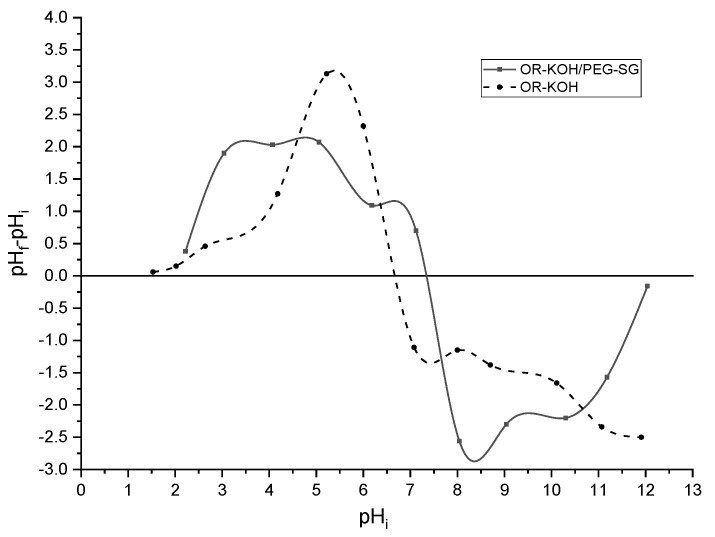
Zero charge points of adsorbents.

**Figure 2 polymers-14-02396-f002:**
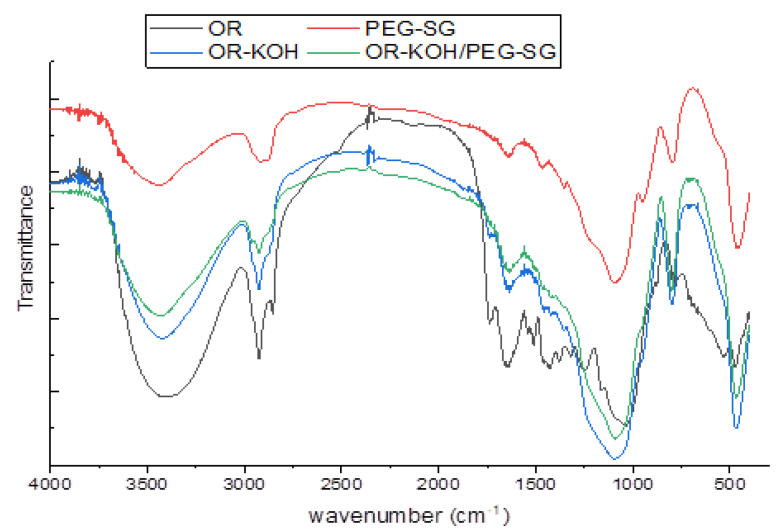
IR spectra of OR, OR-KOH, PEG-SG, and OR-KOH/PEG-SG.

**Figure 3 polymers-14-02396-f003:**
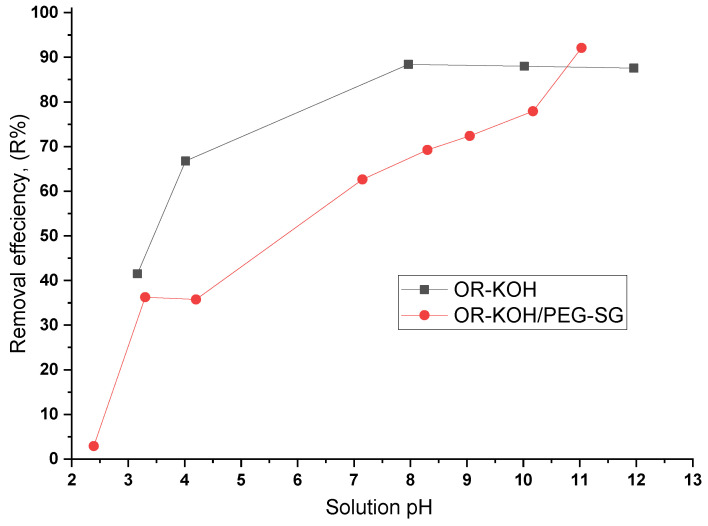
Effect of pH on MB adsorption onto OR-KOH and OR-KOH/PEG-SG adsorbents.

**Figure 4 polymers-14-02396-f004:**
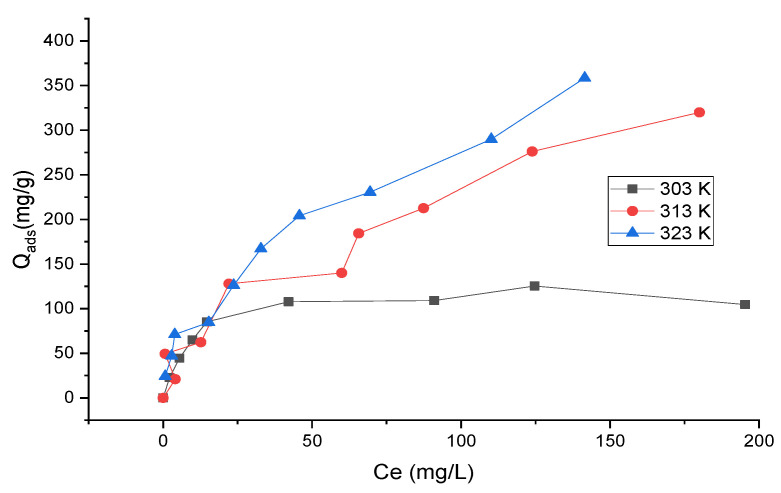
Adsorption isotherms of MB on OR-KOH adsorbent as a function of temperature.

**Figure 5 polymers-14-02396-f005:**
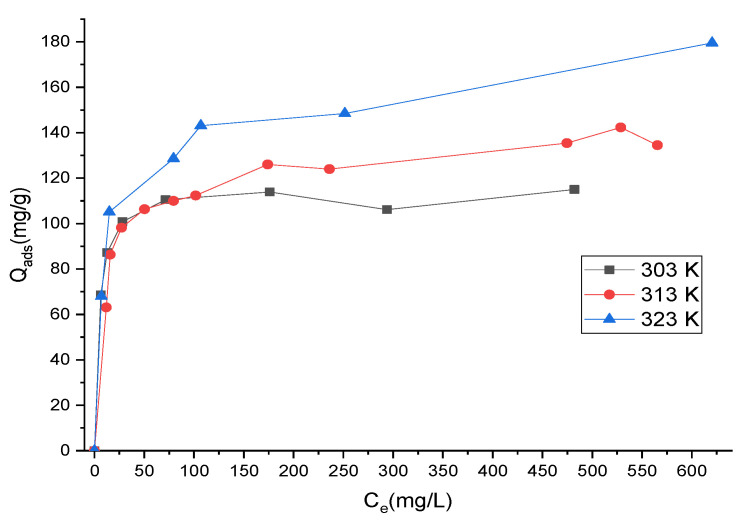
Adsorption isotherms of MB on OR-KOH/PEG-SG adsorbent as a function of temperature.

**Figure 6 polymers-14-02396-f006:**
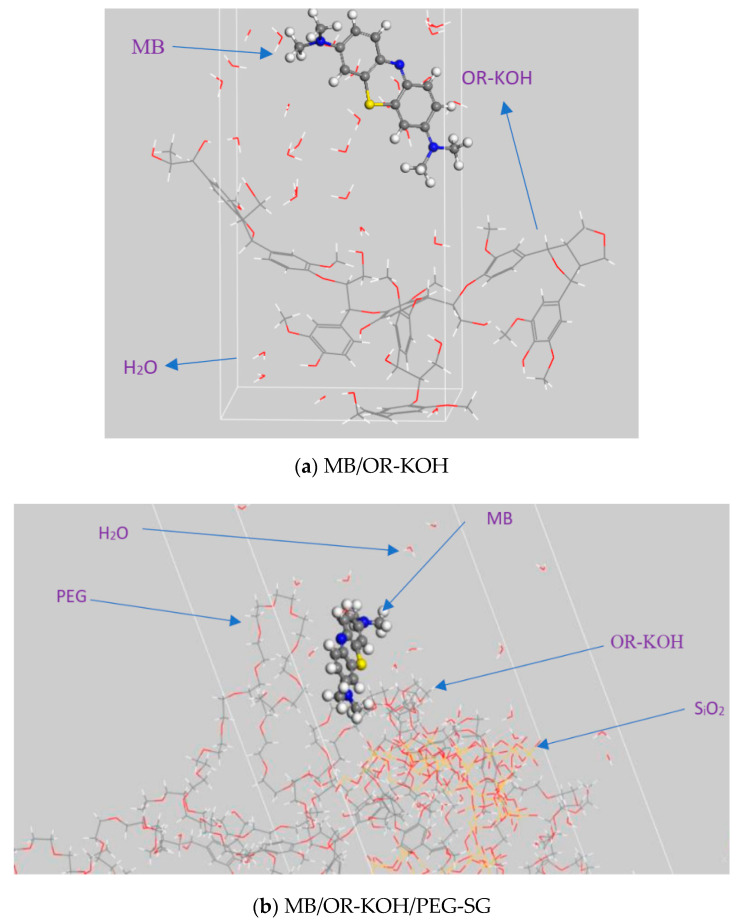
Adsorption of MB in aqueous solution on different adsorbents: (**a**) OR-KOH and (**b**) OR-KOH/PEG-SG.

**Figure 7 polymers-14-02396-f007:**
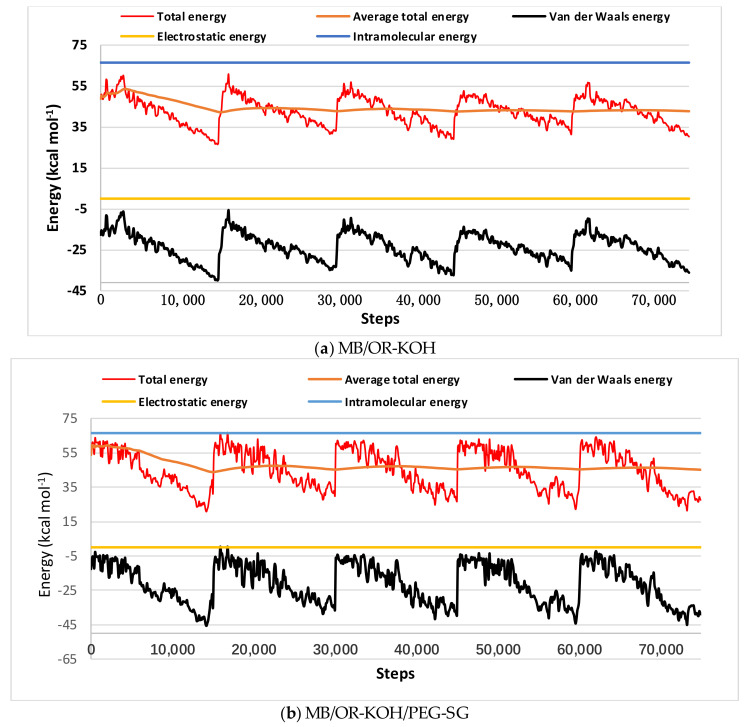
Fluctuating curves of the interaction energies of MB in aqueous solution on different adsorbents: (**a**) OR-KOH and (**b**) OR-KOH/PEG-SG.

**Table 1 polymers-14-02396-t001:** Langmuir and Freundlich parameters for MB adsorption onto bioadsorbent.

OR-KOH								
	Langmuir				Freundlich			
	*q_max_*	*K_L_*	R^2^	RMSE	*K_f_*	*n*	R^2^	RMSE
30°	122.22 ± 5.44	0.1229 ± 0.023	0.97418	8.065	37.218 ± 9.12	4.296 ± 1.04	0.79652	17.45
40°	504.9 ± 40.50	0.0096 ± 0.0040	0.94522	28.84	17.16 ± 5.49	1.77 ± 0.21	0.95474	23.41
50°	492.60 ± 63.78	0.015 ± 0.004	0.96697	23.38	25.013 ± 3.51	1.88 ± 0.11	0.98561	14.01
**OR-KOH/PEG-SG**								
30°	114.28 ± 1.71	0.2462 ± 0.026	0.99430	3.161	69.038 ± 7.18	11.355 ± 2.81	0.74946	8.46
40°	135.09 ± 3.055	0.084 ± 0.011	0.97626	6.39	56.8162 ± 4.69	6.935 ± 0.758	0.90406	7.32
50°	161.44 ± 7.85	0.1028 ± 0.027	0.96388	12.50	59.077 ± 6.83	5.73 ± 0.73	0.93551	9.78

**Table 2 polymers-14-02396-t002:** Adsorption capacities of OR-KOH, OR-KOH/PEG-SG, and other adsorbents.

Adsorbents	q_max_ (mg/g)	Ref.
**PEG-Crosslinked β-CD polymers**	15	[19]
**Chitosan/laterite/iron oxide**	16	[40]
**Iron oxide coated biochar** **nanocomposite (Fe_3_O_4_-BC)**	62.1	[39]
**Silica–PEG intercalated clay hybrids**	98.42	[20]
**Banana stem activated carbon**	101.01	[38]
**OR-KOH/PEG-SG**	161.44	This study
**Cu-doped BTC (Cellulosic woven waste)**	197.90	[41]
**Ba/Co@PEG Nanocomposite**	215.08	[18]
**Sulfonic acid functionalized** **mesoporous silica [S-MSNs]-S[Na]**	224.43	[42]
**Aminated polyacrylonitrile (AMPAN)**	227.2	[43]
**Activated carbon derived** **from sucrose and melamine (ACS)**	454.57	[44]
**Corncob-activated carbon (AHC-KOH)**	489.560	[45]
**OR-KOH**	504.9	This study

**Table 3 polymers-14-02396-t003:** Adsorption energies (kcal/mol) of MB in aqueous solution on OR-KOH and OR-KOH/PEG-SG adsorbents.

Adsorbent	Total Energy	Adsorption Energy	Rigid Adsorption Energy	Deformation Energy	$\mathrm{MB}\;\frac{dE_{ad}}{dN_{i}}$	$H_{2}O\;\frac{dE_{ad}}{dN_{i}}$
**OR-KOH**	53.854	−12.587	−13.550	0.963	−14.998	1.504
**OR-KOH-PEG-SG**	10.913	−55.529	−56.051	0.523	−24.565	0.516

## Data Availability

The data presented in this study are available on request from the corresponding author.

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
