# Peer review of "The Removal of a Textile Dye from an Aqueous Solution Using a Biocomposite Adsorbent"

_polymers, 2022, doi:10.3390/polym14122396_

Round 1

Reviewer 1 Report

This work examines the adsorption of the MB dye on adsorbers prepared from olive mill wastes and their modification. The topic belongs to polymers; however the work needs major revision before publication

Specific points:

  • The authors must highlight the novelty of the work since there are several published papers about the adsorption of dyes from agroindustrial residuals
  • Fig 7 axes : please change the Chinese fonts with English for x axes for example
  • It is not clear why the authors tried to activate the olive residuals with a base and not with acid for example
  • It would be helpful if the authors could add a sem image from the virgin and modified material
  • BET of the materials and correlation with the results?
  • Comparison with other adsorbers? There are hundreds of papers for the removal of dyes available….
  • Can the authors regenerate the adsorbent and reuse it?

Author Response

Reviewer 1

  • The authors must highlight the novelty of the work since there are several published papers about the adsorption of dyes from agroindustrial residuals

Highlights:

We thank the reviewer for this excellent remark. We have accordingly corrected and highlighted the fact that we have used an Olive waste activated with KOH (OR-KOH) and activated and mixed with PEG/silica gel composite (OR-KOH/PEG-SG). The (OR-KOH) and OR-KOH/PEG-SG) were tested as an effective, low-cost adsorbents for methylene blue removal in aqueous solutions:

"A novel adsorbent (OR-KOH/PEG-SG sample) was created by treating olive waste (residue) with KOH and then mixing it with a PEG/silica gel composite. In order to remove methylene blue (MB) from water, these two types of adsorbents were examined in the presence of varying temperatures, pH, and starting adsorbate concentrations. The experimental findings were interpreted and supported by molecular simulations using a classical modeling approach. Molecular simulations were used in conjunction with basic data analysis to better study the adsorption process. Furthermore, Grand Canonical Monte-Carlo (GCMC) atomistic simulations were performed to investigate the microscopic parameters that led to dye binding to the two adsorbents. Finally, a comparison of the obtained data is performed in order to gauge the usefulness of this new approach for doing research."

  • Fig 7 axes: please change the Chinese fonts with English for x-axes, for example

Thank you, we have corrected it

  • It is not clear why the authors tried to activate the olive residuals with a base and not with acid, for example

KOH chemical activation has been reported as a prominent activation method for preparing highly porous activated carbon from different raw materials since it can simultaneously obtain more active sites and better pore structure.

  • It would be helpful if the authors could add an SEM image from the virgin and modified material BET of the materials and correlation with the results?

We completely agree with the reviewer, and frankly, these two characterizations can considerably contribute to the comprehension and depth of our work. To do this, and given the time given by the editor and which was initially 10 days, we asked for an extension of 20 days to carry them out. Unfortunately, this period is very busy with master's and doctoral work. We had the misfortune to know that the equipment broke down, which led us to seek help from another university, but unfortunately, they told us that we had to wait until July or even September. In summary, we have not been able to perform material characterization by SEM and BET in this time frame, and we are very sorry about that.

  • Comparison with other adsorbents? There are hundreds of papers for the removal of dyes available.

A comparison of the adsorption capacity of OR-KOH and OR-KOH/PEG-SG with other adsorbents is added in the text (section 2.4 Adsorption isotherm).

  • Can the authors regenerate the adsorbent and reuse it?

The study of the regeneration of adsorbents by different methods is in progress since the Monte Carlo computing study has shown that the regeneration of the adsorbent is possible.  

Reviewer 2 Report

Dear Authors

An essential point regarding the batch adsorption experiments need to be declared first before your work can be considered further for publication.

The point is: you did not indicate if the adsorption experiments performed under stirring or not?

Without stirring, your adsorption experiments were performed under different contact time and not all of the adsorbents particles were involved for the same time in the adsorption process.

Accordingly, all the obtained results are obtained under different "actual"contact time even for the same experiments performed under the same experimental contact time.   

Author Response

Reviewer 2

An essential point regarding the batch adsorption experiments need to be declared first before your work can be considered further for publication. The point is: you did not indicate if the adsorption experiments performed under stirring or not? Without stirring, your adsorption experiments were performed under different contact time and not all of the adsorbents particles were involved for the same time in the adsorption process. Accordingly, all the obtained results are obtained under different "actual" contact time even for the same experiments performed under the same experimental contact time. 

Please accept our gratitude for this insightful and well-received comment from the reviewer. A section 1.2 error has been made with regard to the batch adsorption tests; we tell him that they were performed under stirring at a constant temperature.

Round 2

Reviewer 1 Report

the authors revised their work, therefore the paper can proceed for publication

Author Response

Responses to Reviewer 1

The paper was improved as required by the two reviewers

Reviewer 2 Report

Dear Authors

After screening of your revised manuscript, I can recommend for publication after major revision. The comments are mentioned below.

Abstract: The best findings especially the maximum adsorption capacity should be mentioned.

Introduction: The authors did not mentioned any thing about the use of PEG/silica gel composite for dyes removal and water treatment in general. This issue must be covered in the introduction to declare the contribution of the PEG/silica gel composite in the removal process of MB dye. Many published results can be cited and used for comparison with current work performed by the authors.

Materials and methods

The authors performed the adsorption experiments using the developed OR-KOH and OR-KOH/PEG-SG composite adsorbents and performed a comparison between the obtained results. Why the authors did not perform the same adsorption experiments for PEG-SG composite ?? to compare with as a parent matrix component just like the OR-KOH adsorbent component.

This experiments should be performed and the obtained results should be discussed to show the synergetic effect between OR-KOH adsorbent component and the PEG-SG composite component.

starting after section 1.2. Chemicals, all other section have the same section number. This should be corrected. 

Results and Discussion

All the performed characterization and adsorption experiments and their characterization should be included the results of the PEG-SG composite component and rewrite accordingly.   

Author Response

Reponses to Reviewer 2

  1. Abstract: The best findings especially the maximum adsorption capacity should be mentioned.

Response: Thank you for your pertinent inquiry. In the amended paper, we included a comprehensive comparison of many adsorbents (Table 2). The text that was added is marked in yellow.

  1. Introduction: The authors did not mention any thing about the use of PEG/silica gel composite for dyes removal and water treatment in general. This issue must be covered in the introduction to declare the contribution of the PEG/silica gel composite in the removal process of MB dye. Many published results can be cited and used for comparison with current work performed by the authors.

 Response:  The required discussion in the introduction was added.

  1. Materials and methods

The authors performed the adsorption experiments using the developed OR-KOH and OR-KOH/PEG-SG composite adsorbents and performed a comparison between the obtained results. Why the authors did not perform the same adsorption experiments for PEG-SG composite ?? to compare with as a parent matrix component just like the OR-KOH adsorbent component.

These experiments should be performed, and the obtained results should be discussed to show the synergetic effect between OR-KOH adsorbent component and the PEG-SG composite component.

  1. Starting after section 1.2. Chemicals, and all other section have the same section number. This should be corrected.

Response: Corrected

  1. Results and Discussion

All the performed characterization and adsorption experiments and their characterization should be included in the results of the PEG-SG composite component and rewritten accordingly. 

Response to questions 3 and 4: This research attempts to use olive waste as an inexpensive dye-removal adsorbent. The support was chemically activated using KOH and then encapsulated with a PEG-SG composite. The primary function of encapsulation is to immobilize the adsorbent for industrial applications, hence improving the solid/liquid separation of treated effluent. The findings indicate that the activated olive waste OR-KOH is an excellent biosorbent for MB. It may be encapsulated with PEG-SG composite to produce the OR-KOH/PEG-SG material, which also has strong adsorption characteristics toward MB.
